# Brief communication: Unabated wastage of the Juneau and Stikine icefields (southeast Alaska) in the early twenty-first century

Etienne BERTHIER[1], Christopher LARSEN[2], William J. DURKIN[3], Michael J. WILLIS[4], Matthew E. PRITCHARD[3]

[1] LEGOS, Université de Toulouse, CNES, CNRS, IRD, UPS, F-31400 Toulouse, France
[2] Geophysical Institute, University of Alaska Fairbanks, Fairbanks, Alaska, USA
[3] Earth and Atmospheric Sciences Department, Cornell University, Ithaca, New York, USA
[4] Cooperative Institute for Research in Environmental Sciences (CIRES), University of Colorado, Boulder, CO, USA

*Correspondence to*: Etienne Berthier (etienne.berthier@legos.obs-mip.fr)

**Abstract.** The large Juneau and Stikine icefields (Alaska) lost mass rapidly in the second part of the 20th century. Laser altimetry, gravimetry and field measurements suggest continuing mass loss in the early 21st century. However, two recent studies based on time series of SRTM and ASTER digital elevation models (DEMs) indicate a slowdown in mass loss after 2000. Here, the ASTER-based geodetic mass balances are recalculated carefully avoiding the use of the SRTM DEM because of the unknown penetration depth of the C-Band radar signal. We find strongly negative mass balances from 2000 to 2016 (-0.68±0.15 m w.e. a$^{-1}$ for the Juneau Icefield and -0.83±0.12 m w.e. a$^{-1}$ for the Stikine Icefield), in agreement with laser altimetry, confirming that mass losses are continuing at unabated rates for both icefields. The SRTM DEM should be avoided or used very cautiously to estimate glacier volume change, especially in the North Hemisphere and over timescales of less than ~20 yrs.

## 1 Introduction

The Juneau Icefield (JIF) and Stikine Icefield (SIF) are the southernmost large icefields in Alaska (Figure 1). The JIF covers about 3800 km² and the SIF close to 6000 km² at the border between southeast Alaska and Canada (Kienholz et al., 2015). Together they account for roughly 10% of the total glacierized area in Alaska. Both icefields experienced rapid mass loss in the second part of the 20th century (Arendt et al., 2002; Berthier et al., 2010; Larsen et al., 2007). Spaceborne gravimetry and laser altimetry data indicate continuing rapid mass loss in southeast Alaska between 2003 and 2009 (Arendt et al., 2013).

For the JIF, Larsen et al. (2007) found a negative mass balance of -0.62 m w.e. a$^{-1}$ for a time interval starting in 1948/1982/1987 (depending on the map dates) and ending in 2000, the date of acquisition of the shuttle radar topographic mission (SRTM) digital elevation model (DEM). Berthier et al. (2010) found a slightly less negative multi-decadal mass balance (-0.53 ± 0.15 m w.e. a$^{-1}$) from the same starting dates as Larsen et al. (2007) to a final DEM acquired in 2007. Repeat airborne laser altimetry are available for nine glaciers of the JIF (Larsen et

al., 2015) with a first survey performed in 1993 (2 glaciers), 1999 (1 glacier) and 2007 (6 glaciers). The last survey used in Larsen et al. (2015) was flown in 2012 for all glaciers. During these varying time intervals, nine glaciers experienced strongly negative mass balances (between -0.51 and -1.14 m w.e. a$^{-1}$) while Taku Glacier, which alone accounts for one fifth of the JIF area, experienced a slightly positive mass balance (+0.13 m w.e. a$^{-1}$). Further, the glaciological measurements performed on Lemon Creek Glacier, a world glacier monitoring service (WGMS) reference glacier covering 11.8 km² in 1998, suggest accelerated mass loss since the mid-1980s: the glacier-wide mass balance declined from -0.30 m w.e. a$^{-1}$ between 1953 and 1985 to -0.60 m w.e. a$^{-1}$ between 1986 and 2011 (Pelto et al., 2013). The trend toward enhanced mass loss is also observed on Taku Glacier, for which the mass balance was positive (+0.42 m w.e. a$^{-1}$) from 1946 to 1988 and negative (-0.14 m w.e. a$^{-1}$) from 1988 to 2006 (Pelto et al., 2008). A modelling study also found a negative mass balance for the entire JIF (-0.33 m w.e. a$^{-1}$) for 1971-2010 (Ziemen et al., 2016). Their 40-year mass balance is a result of glacier mass stability until 1996 and rapid mass loss afterwards. Taken together, all these studies point toward rapid mass loss of the JIF and accelerated wastage during the last ~20 years. Conversely, a study based on the SRTM DEM and Advanced Spaceborne Thermal Emission and Reflection Radiometer (ASTER) multi-temporal DEMs found a JIF mass balance only moderately negative at -0.13 ± 0.12  m w.e. a$^{-1}$ from 2000 to 2009/2013 (Melkonian et al., 2014).

Only a few estimates of mass change are available on the larger and more remote SIF. Three of its glaciers were surveyed with airborne laser altimetry from 1996 to 2013 and all experienced rapid mass loss (Larsen et al., 2015). The glacier-wide mass balances were -0.71, -0.98 and -1.19 m w.e. a$^{-1}$ for, respectively, Baird, Le Conte and Triumph glaciers (Figure 1). Based on DEM differencing over several decades, Larsen et al. (2007) and Berthier et al. (2010) found SIF-wide mass balance of, respectively, -1.48 and -0.76 ± 0.12 m w.e. a$^{-1}$. A recent estimate based on the SRTM and ASTER DEMs suggest a less negative icefield-wide mass balance of -0.57 ± 0.18 m w.e. a$^{-1}$ from 2000 to 2014 (Melkonian et al., 2016).

If correct, Melkonian et al. (2014, 2016)'s estimates would imply a considerable slowdown of the mass loss of the Juneau and, to a smaller extent, Stikine icefields during the first decade of the 21$^{st}$ century. However, no clear trend in climate such as cooling or increased precipitation was found during this period to explain such a slowdown (Melkonian et al., 2014; Ziemen et al., 2016). Field observations of the equilibrium line altitudes and surface mass balances on Lemon Creek and Taku glaciers (JIF) also do not support a slowdown (WGMS, 2017). Melkonian et al. (2014, 2016)'s estimates used as starting elevation measurement the C-Band SRTM DEM acquired in February 2000, the core of winter in Alaska. The C-Band radar signal is known to penetrate into the cold winter snow and firn such that SRTM maps a surface below the real glacier surface which can bias the elevation change measurements (e.g., Berthier et al., 2006; Rignot et al., 2001). Melkonian et al. (2014, 2016) accounted for this penetration by subtracting the simultaneous C-Band and X-Band SRTM DEMs, assuming no penetration of the X-Band DEM (Gardelle et al., 2012), the best available correction at the time of their study. However, this strategy may not be appropriate given that the X-band penetration depth has recently been

recognized to reach several meters in cold and dry snow/firn (e.g., Dehecq et al., 2016; Round et al., 2017). In this context, the goal of this brief communication is to recalculate the early 21$^{st}$ century geodetic mass balances of the Juneau and Stikine icefields using multi-temporal ASTER DEMs, carefully excluding the SRTM DEM to avoid a likely penetration bias.

## 2 Data, methods and uncertainties

The data and methodology applied to the JIF and SIF were identical to the ones used in a recent study deriving region-wide glacier mass balances in High Mountain Asia (Brun et al., 2017). The reader is thus referred to the latter study for details. Only the main processing steps are briefly presented here.

ASTER DEMs were calculated using the open-source Ames Stereo Pipeline (ASP) (Shean et al., 2016) from 3N (nadir) and 3B (backward) images acquired between 2000 and 2016. Images with cloud coverage lower than 80% were selected, resulting in 153 stereo pairs for the JIF and 368 stereo pairs for the SIF. DEMs in which valid elevation data covered less than 0.5% of the icefield areas were excluded, reducing the number of DEMs to 114 for the JIF and 284 for the SIF. Planimetric and altimetric offsets of each ASTER DEM were corrected using the SRTM DEM as a reference (Nuth and Kääb, 2011). Offsets were determined on stable terrain, masking out glacierized areas using the Randolph Glacier Inventory v5.0 (Pfeffer et al., 2014). The RGI v5.0 glacier outlines for both the JIF and SIF were mapped using imagery acquired in majority in August of 2004 and 2005 (Bolch et al., 2010; Kienholz et al., 2015). No updated inventory is available or was produced during this study for the JIF and SIF. Therefore, we neglected changes in glacierized area between 2000 and 2016, and assumed that mass balance uncertainties linked to area changes are covered by our 5% area uncertainty (Paul et al., 2013, Dussaillant et al., 2018).

For the JIF only, we also downloaded directly the ASTER DEMs available online from the LPDAAC website (called AST14DEM) because they were used in Melkonian et al. (2014, 2016). The goal is to test the sensitivity of the JIF-wide mass balance to the ASTER DEM generation software. 3D coregistration of the AST14DEMs was performed using the same steps as the ASP DEMs. Unlike the ASP DEMs, the AST14DEMs contain no data gaps, as they are filled by interpolation.

From the time series of 3D-coregistered ASTER DEMs, the rate of elevation changes (*dh/dt* in the following) was extracted for each pixel of our study domain in two steps (Berthier et al., 2016). The SRTM DEM was excluded when extracting the final *dh/dt*. *dh/dt* were calculated for the entire period (from 2000 to 2016) and also for different sub-periods for the sake of comparability to published mass balance estimates.

For both icefields and in each 50-m altitude interval, *dh/dt* lying outside of ±3 normalized median absolute deviations (NMAD) were considered as outliers. We further excluded all *dh/dt* measurements for which the

error in the linear fit is larger than 2 m a$^{-1}$. The total volume change rate was calculated as the integral of the mean *dh/dt* over the area altitude distribution. The icefield-wide mass balances were obtained using a volume-to-mass conversion factor of 850 kg m$^{-3}$ (Huss, 2013). The same procedure was followed to compute the glacier-wide mass balances of selected glaciers for which mass balances were estimated from repeat laser altimetry surveys (Larsen et al., 2015).

Uncertainties for *dh/dt* were computed using a method which consists in splitting the off-glacier terrain in 4 by 4 tiles (Berthier et al., 2016). For each tile, the mean *dh/dt* off-glacier is computed. The uncertainty is then calculated as the mean absolute difference for these 16 tiles. We found uncertainties of 0.03 m a$^{-1}$ for JIF and 0.04 m a$^{-1}$ for SIF from 2000 to 2016. When data gaps occurred in the *dh/dt* map, we conservatively multiplied these uncertainties by a factor of five. A ± 5% uncertainty for glacier area (Paul et al., 2013) and ± 60 kg m$^{-3}$ for the density conversion factor (Huss, 2013) were used.

## 3 Results

Rate of elevation changes for the two icefields from 2000 to 2016 are mapped in Figure 1. Most glaciers thinned rapidly in their lower parts and experienced limited elevation change in their upper reaches. Thinning rates as negative as 9 m a$^{-1}$ are observed. Taku Glacier (southern outlet of the JIF) is an exception with thickening of up to 4 m a$^{-1}$ at its glacier front. Understanding the pattern of *dh/dt* and its variability among glaciers is beyond the scope of this brief communication and the reader is referred to earlier publications on this topic (e.g., Larsen et al., 2015).

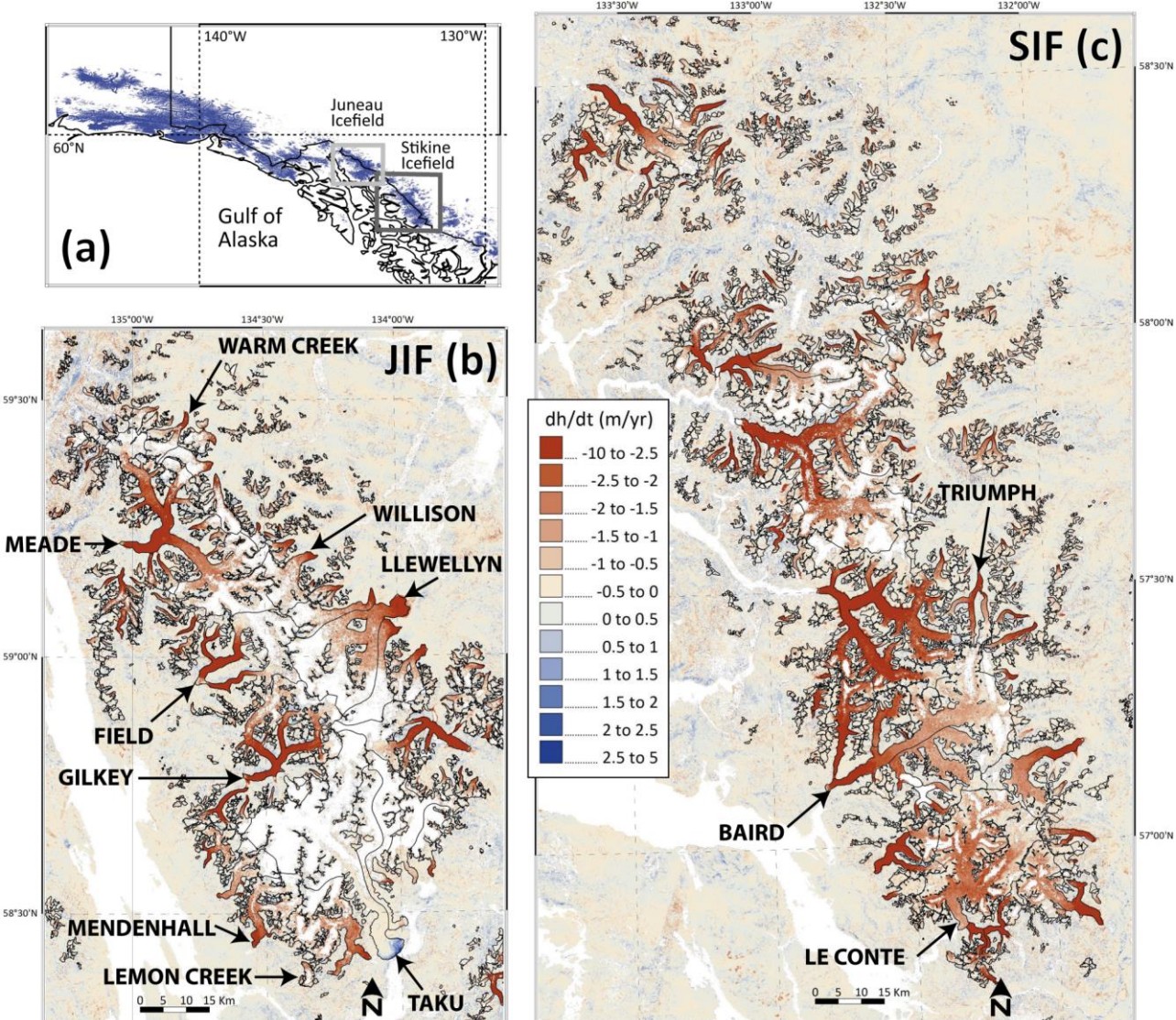

**Figure 1:** Rate of elevation changes for the Juneau and Stikine icefields from 2000 to 2016. (a) Location of the two icefields in southeast Alaska. Rate of elevation changes (*dh/dt*) for the JIF (b) and (c) for the SIF. Glacier outlines are from RGI v5.0. Glaciers surveyed by airborne laser altimetry are labelled. The horizontal scale and the color code are the same for the two maps. Areas in white correspond to data gaps.

The 2000-2016 mass balances are clearly negative for both icefields at -0.68±0.15 m w.e. a$^{-1}$ for JIF (59% coverage with valid data) and -0.83±0.12 m w.e. a$^{-1}$ for SIF (81% coverage with valid data). Our values are 0.51±0.18 m w.e. a$^{-1}$ (JIF) and 0.21±0.25 m w.e. a$^{-1}$ (SIF) more negative than in Melkonian et al. (2014, 2016) and statistically different for the JIF, i.e. the JIF mass balances do not overlap given the error bars. If we apply the linear regression analysis to a subset of the ASTER DEMs to match the time periods studied by Melkonian et al. (2014, 2016), the icefield-wide mass balances remain mostly unchanged: -0.64±0.14 m w.e. a$^{-1}$ for JIF from 2000 to 2013, 44% coverage with valid data; -0.78±0.17 m w.e. a$^{-1}$ for SIF from 2000 to 2014, 55% coverage with valid data.

The coverage with valid *dh/dt* data drops rapidly for both icefields when shorter time periods are considered, especially at high elevation. For example, the percentage of valid data is reduced to only 8% (respectively 25%) on the JIF when the 2000-2008 (respectively 2008-2016) period is analyzed. Thus, the ASTER multi-temporal

analysis is not appropriate to measure mass balance over periods shorter than 10 years for these two Alaskan icefields. This is due to the presence of many cloudy images and, for cloud-free scenes, to a large percentage of data gaps in individual ASTER DEMs over the accumulation areas of the icefields, a direct result of the limited contrast in the ASTER stereo-images over textureless snow fields.

In Figure 2, *dh/dt* are plotted as a function of altitude and compared to the values in Melkonian et al. (2014, 2016). To enable a more direct comparison, we applied the same criteria to average their *dh/dt* in 50-m altitude bands and exclude outliers. We also considered the same periods, from 2000 to 2013 for the JIF and from 2000 to 2014 for the SIF. In the case of the SIF (Figure 2b), we also added the *dh/dt* obtained by applying our method to the AST14DEMs.

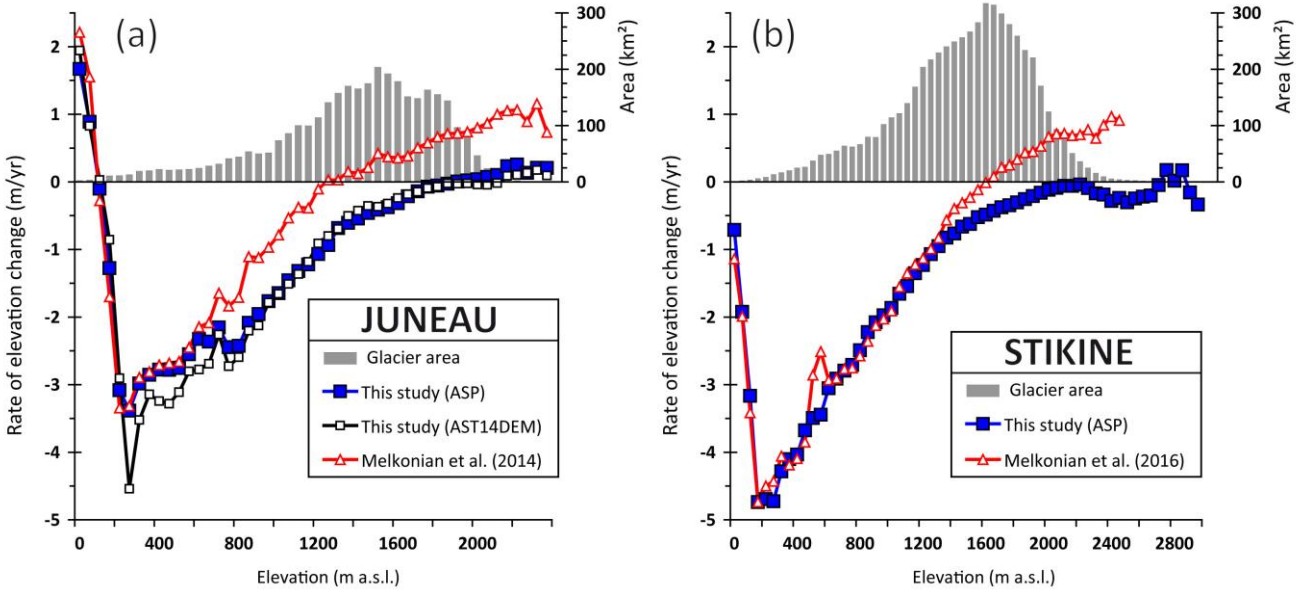

**Figure 2:** Rates of elevation change vs. elevation for the JIF from 2000 to 2013 (a) and for the SIF from 2000 to 2014 (b). Results from this study are compared to the *dh/dt* values obtained in two earlier studies using a similar method (Melkonian et al., 2014, 2016). The grey histograms show the area-altitude distribution.

For the JIF, an excellent agreement is found between the *dh/dt* values obtained in this study using the ASP DEMs and the AST14DEMs, except between 250 and 600 m a.s.l. (5% of the icefield area) where the thinning rates are about 0.5 m $a^{-1}$ more negative using the AST14DEMs. The area-weighted mean absolute difference between these two curves (ASP and AST14DEM) is 0.09 m $a^{-1}$. The Melkonian et al. (2014)'s *dh/dt* generally agree with ours below 600 m a.s.l. Above this elevation, their values are systematically more positive. The difference reaches 0.7 m $a^{-1}$ at 800 m a.s.l. and then remains more or less stable, around 0.7-0.9 m $a^{-1}$. Melkonian et al. (2014) data suggests thickening of the areas above 1350 m a.s.l. where 62% of the JIF area is located.

For SIF, a good agreement is found between ours and Melkonian et al. (2016)'s *dh/dt* below an elevation of 1300 m a.s.l. Above 1300 m the two curves diverge. Our *dh/dt* are becoming less negative until 2100 m a.s.l.

where they become indistinguishable from 0 m $a^{-1}$ up to the SIF highest elevation band. Conversely, in the Melkonian et al. (2016) dataset, *dh/dt* increases rapidly, crossing 0 m $a^{-1}$ at ~1650 m a.s.l., finally arriving at a thickening rate of > 0.7 m $a^{-1}$ above 2000 m a.s.l. Thus the difference in SIF-wide mass balance between the two datasets is due to difference in *dh/dt* above 1300 m a.s.l., where 66% of the SIF icefield area is found.

Comparison of our *dh/dt* estimates to the ones derived from repeat laser altimetry data is not straightforward because the survey periods differ. For example, for the JIF, six out of nine glaciers were sampled for the first time in 2007. In most cases, it would be technically possible to use a temporal subset of the ASTER DEMs to match the time period of altimetry surveys but, as said above, this would be at the cost of the coverage in our *dh/dt* maps and would lead to much more uncertain mass balance estimates. Consequently, we preferred to extract *dh/dt* and the individual glacier mass balance for the longest available time period in the ASTER series (from 2000 to 2016) in order to maximize coverage and thus minimize uncertainties. A further complication for the comparison of our ASTER-based results to repeat laser altimetry arises from different spatial sampling: mostly continuous coverage from DEMs vs. centreline sampling from laser altimetry. Berthier et al. (2010) found that centreline sampling could lead to an overestimation of mass loss. In their study, two large and rapidly retreating glaciers (Bering and Columbia, outside of our study domain) were responsible for 92% of the overestimation of the mass loss from centreline profiling (Table S4 in Berthier et al., 2010). Overestimation was not obvious for other glaciers. More recently, Johnson et al. (2013) presented an improved treatment of laser altimetry data and found no such overestimation from centerline profiling over the Glacier Bay region (southeast Alaska). In their improved processing, each change in elevation (*dz*) is assigned to a mid-point between old and new elevations whereas in the original laser altimetry analysis (Arendt et al., 2002), *dz* were assigned to the old elevation.

The pattern of *dh/dt* with altitude for individual glaciers is in broad agreement between laser altimetry and our ASTER-based results (Supplementary Figure S1). Importantly, for both datasets, no clear thickening was observed in the accumulation areas of glaciers. When individual elevation bins of 50 m are considered, averaged differences between *dh/dt* from laser altimetry and the ASTER DEMs are typically 0.2 to 0.3 m $a^{-1}$ for individual glaciers. This level of error is similar to the one found previously for the ASTER method in the Mont-Blanc area (Berthier et al., 2016).

Glacier-wide mass balances for individual glaciers match well (Table 1, Supplementary Figure S2). The mean mass balance of these 12 glaciers is nearly the same (-0.73 and -0.74 m w.e. $a^{-1}$) using the two techniques. The standard deviation of the mass balance difference is 0.18 m w.e. $a^{-1}$ (n=12). For 60 individual glaciers larger than 2 km² in High Mountain Asia, Brun et al. (2017) also found a standard deviation of 0.17 m w.e. $a^{-1}$ between the ASTER-based and published glacier-wide mass balance estimates. In the very different geographic context of large maritime glaciers of southeast Alaska, we confirm here their uncertainty estimate for individual glaciers in High Mountain Asia.

209

Our results are also in good agreement with glaciological measurements on Taku and Lemon Creek glaciers. For

Taku Glacier, the mass balance was -0.01 m w.e. a$^{-1}$ between September 2000 and September 2011 (Pelto et al.,

2013) and -0.08 m w.e. a$^{-1}$ between September 2000 and September 2016 (WGMS, 2017). We derived a very

similar glacier-wide mass balance (-0.01 ± 0.16 m w.e. a$^{-1}$) from ASTER DEMs acquired between 2000 and 2016.

Conversely, Melkonian et al. (2014)'s mass balance for Taku Glacier was strongly positive at +0.44 ± 0.15 m w.e.

a$^{-1}$. The 2000-2016 mass balance for Lemon Creek Glacier was -0.56 m w.e. a$^{-1}$ (WGMS, 2017) while our ASTER-

based mass balance is just slightly more negative at -0.78 ± 0.14 m w.e. a$^{-1}$.

**Table 1**. Glacier-wide mass balances (B$_a$) of 12 individual glaciers of the JIF and SIF derived from airborne laser altimetry for

different periods (Larsen et al., 2015) and calculated in this study using ASTER DEMs from 2000 to 2016. Uncertainties for

the mean mass balances of 9 (JIF) and 3 (SIF) and 12 (JIF and SIF) glaciers are calculated as the area-weighed mean of

uncertainties for individual glaciers.

| Icefield/Glacier | Area km² | Laser period | B$_a$ Laser m w.e. a$^{-1}$ (Larsen et al., 2015) | B$_a$ ASTER m w.e. a$^{-1}$ (this study) |
|---|---|---|---|---|
| **Juneau** | **3398** | | | **-0.68 ± 0.15** |
| Field | 187 | 2007-2012 | -0.94 ± 0.26 | -0.93 ± 0.16 |
| Gilkey | 223 | 2007-2012 | -0.75 ± 0.23 | -0.99 ± 0.14 |
| Lemon Creek | 9 | 1993-2012 | -0.91 ± 0.48 | -0.78 ± 0.14 |
| Llewellyn | 435 | 2007-2012 | -0.61 ± 0.15 | -0.70 ± 0.17 |
| Meade | 446 | 2007-2012 | -1.03 ± 0.26 | -0.88 ± 0.15 |
| Mendenhall | 106 | 1999-2012 | -0.57 ± 0.87 | -0.73 ± 0.13 |
| Taku | 711 | 1993-2012 | 0.13 ± 0.10 | -0.01 ± 0.16 |
| Warm Creek | 39 | 2007-2012 | -0.67 ± 0.31 | -0.71 ± 0.16 |
| Willison | 79 | 2007-2012 | -0.51 ± 0.38 | -0.69 ± 0.15 |
| **Sum/Mean 9 glaciers** | **2234** | | **-0.65 ± 0.22** | **-0.71 ± 0.16** |
| | | | | |
| **Stikine** | **5805** | | | **-0.83 ± 0.12** |
| LeConte | 56 | 1996-2013 | -0.98 ± 0.31 | -0.93 ± 0.13 |
| Baird | 435 | 1996-2013 | -0.71 ± 0.12 | -0.70 ± 0.12 |
| Triumph | 356 | 1996-2013 | -1.19 ± 0.48 | -0.86 ± 0.10 |
| **Sum/Mean 3 glaciers** | **847** | | **-0.96 ± 0.28** | **-0.83 ± 0.12** |
| | | | | |
| **Mean all 12 Glaciers** | | | **-0.73 ± 0.24** | **-0.74 ± 0.15** |

## 4 Discussion

We find an excellent agreement between repeat laser altimetry survey and our multi-temporal analysis of ASTER DEMs both in term of mass balances and pattern of *dh/dt* with altitude for the JIF and SIF since 2000 (Supplementary Figure S1-S2). This agreement suggests that an appropriate analysis of centreline data may be sufficient to measure the glacier-wide mass balance of these glaciers as previously shown for the nearby Glacier Bay area (Johnson et al., 2013). Our results also suggest that the limited number of glaciers sampled using laser altimetry are representative of the icefields as a whole. This is rather expected for the JIF because 9 glaciers covering a large fraction of the icefield (66%) were monitored using airborne data but not straightforward for the SIF where only 3 glaciers, accounting for 15% of the total icefield area, were surveyed.

This agreement between our ASTER results and airborne laser altimetry, together with the fact that most studies point toward steady or accelerating mass losses in southeast Alaska (see introduction), suggest that the mass balance is overestimated in Melkonian et al. (2014, 2016). There are two main differences between Melkonian et al. (2014, 2016)'s method and ours that could explain these contending mass balances: (i) they did not generate the DEM themselves but directly download the AST14DEM product from the LPDAAC website and (ii) they used the SRTM DEM as a starting elevation in their regression analysis to compute *dh/dt*.

To test the sensitivity of our results to the ASTER DEM generation software, we applied our processing chain (in particular, excluding the SRTM DEM to infer the final *dh/dt*) to the AST14DEMs. From 2000 to 2016, we found a JIF-wide mass balance of -0.67±0.27 m w.e. a$^{-1}$, in striking agreement with the value derived from ASP DEMs (-0.68±0.15 m w.e. a$^{-1}$). The pattern of *dh/dt* with elevation is also in excellent agreement (Figure 2a). Uncertainties are nearly doubled when applying our method to the AST14DEMs: this is explained by larger errors of *dh/dt* off glacier (0.06 m a$^{-1}$ for AST14DEMs vs. 0.03 m a$^{-1}$ for ASP DEMs) and a lower coverage of the JIF with valid *dh/dt* data (49% for AST14DEMs vs. 59% for ASP DEMs). The latter may appear counter-intuitive as the AST14DEMs are delivered with no data gaps. The larger percentage of data gaps in the final AST14DEMs *dh/dt* maps results from the higher noise level of the individual AST14DEMs and demonstrate the efficiency of our filters to exclude unreliable *dh/dt* values.

Thus, we conclude that Melkonian et al. (2014, 2016) found too positive mass balance for the JIF and, to a lesser extent, for the SIF because of the penetration of the SRTM C-Band radar signal into cold winter snow and firn. This interpretation is further supported by the fact that *dh/dt* curves nicely agree in the ablation areas where SRTM penetration depth is negligible and diverge in the colder and drier accumulation areas where larger penetration depths are expected (Figure 2). As noted in the introduction, Melkonian et al. (2014, 2016) accounted for this by subtracting the C-Band and X-Band SRTM DEM, assuming no penetration of the X-Band DEM (Gardelle et al., 2012). However, X-band penetration can reach several meters into cold snow and firn

(e.g., Dehecq et al., 2016; Round et al., 2017). In the case of the SIF, Melkonian et al. (2016) assumed no penetration below 1000 m a.s.l. and 2 m for elevations above 1000 m. Aware of how uncertain this correction was, these authors also proposed (their supplementary material section 6.3 and, Table S4) a different correction with no penetration below 1000 m a.s.l. and a linear increase from 2 to 8 m from 1000-2500 m a.s.l. Using this alternative scenario, they found an icefield-wide mass balance of -0.85 m w.e. a$^{-1}$, in better agreement with our value of -0.78±0.17 m w.e. a$^{-1}$ from 2000 to 2014. Their 2 to 8 m penetration depth is consistent with the penetration gradient we inferred here by subtracting the SRTM DEM from a reconstructed DEM, obtained by extrapolating *dh/dt* to the time of acquisition of the SRTM as proposed in Wang and Kääb (2015). This is also consistent with a first-order estimate of the penetration depth inferred from the elevation difference between the SRTM DEM and laser altimetry profiles acquired in late August 1999 and May 2000 over Baird and Taku glaciers. However, the latter estimates should be considered with care given the complexity to account simultaneously for seasonal elevation changes, long term elevation changes and the difficulty to estimate the vertical offset between the two elevation datasets on ice-free terrain.

The fact that the positive bias in Melkonian et al. (2014, 2016) mass balances was larger for the JIF than for the SIF suggests a larger SRTM penetration depth for the JIF. It indicates that this penetration is probably spatially variable (depending on the firn conditions in February 2000) such that a correction determined on a single icefield (or worse a single glacier) may not apply to neighbouring glacier areas.

Larsen et al. (2007) used the SRTM DEM as their final topography after applying a linear correction of SRTM with altitude (2.6 m per 1000 m elevation, with a -2.5 offset at 0 elevation) determined by comparing SRTM to August 2000 laser altimetry data. Such a correction would correspond to a maximum SRTM penetration of ~1.5-2 m above 1500 m a.s.l., much smaller than what we found here. Thus, the fact that SRTM penetration depth is larger than previously thought over southeast Alaska icefields may explain why Larsen et al. (2007) found larger mass losses than Arendt et al. (2002) and Berthier et al. (2010) who both used only non-penetrating optical data (lidar or stereo-imagery).

An uneven seasonal distribution of the ASTER DEMs could bias the multi-annual mass balances derived using the ASTER method (Berthier et al., 2016). This is especially crucial in maritime environment such as southeast Alaska where large seasonal height variations are expected. As in the case of the Mont-Blanc area (Figure 6 in Berthier et al., 2016), we sampled an hypothetic seasonal cycle in surface elevation changes at the time of acquisition of all ASTER DEMs over the JIF and fitted a linear regression to the elevation change time series. Assuming a seasonal amplitude as large as 10 m (a value in agreement with field measurements of the Juneau Icefield Mass Balance Program, Pelto et al., 2013), the slope of the regression line is very close to 0 (-0.007 m a$^{-1}$) suggesting no seasonal bias in the dates of the ASTER DEMs. To confirm the lack of seasonal bias and because the majority of the ASTER images were acquired close to accumulation peak, we also calculated a mass balance for the JIF considering only the 61 ASTER DEMs acquired in March, April and May between 2000 and 2016. For

this alternative mass balance estimate, the coverage with valid data is reduced to 38%. At -0.58±0.18 m w.e. a$^{-1}$, the JIF-wide mass balance is slightly less negative but not statistically different from the "all seasons" value (-0.68±0.15 m w.e. a$^{-1}$, 59% of valid data). The pattern of *dh/dt* with altitude is also very similar.

## 5 Conclusion

Our ASTER-based analysis shows that the Juneau and Stikine icefields continued to lose mass rapidly from 2000 to 2016, a finding in agreement with the repeat laser altimetry and field based measurements. The mass balances from repeat airborne laser altimetry and multi-temporal ASTER DEMs are reconciled if the SRTM DEM is discarded when extracting the rate of elevation change on glaciers from the elevation time series. Multi-temporal analysis of DEMs derived from medium resolution satellite optical stereo-imagery is thus a powerful method to estimate geodetic region-wide mass balances over time intervals of, typically, more than 10 years. Shorter time intervals can now be measured using very high resolution imagery (e.g., Worldview and Pléiades). The strength of the ASTER method lies in the fact that it is based on an homogeneous and continuous archive of imagery built since 2000 using the same sensor. Maintaining openly available medium- to high-resolution stereo capabilities should be a high priority among space agencies in the future.

Previously published mass balances for these Alaska icefields using SRTM and ASTER DEMs were likely biased positively because of the strong penetration of the C-Band and X-Band radar signal into the cold winter snow and firn in February, when the SRTM was flown. Accounting for this penetration by subtracting the C-Band and X-Band SRTM DEMs (as often done before) is not appropriate because the X-Band penetration depth can also sometimes reach several meters if radar images are acquired under cold and dry conditions. Under wet conditions, when water is present in the snow and firn upper layers, this penetration is reduced. Even so, caution should thus be used when deriving mass balance using SRTM and Tandem-X DEMs over time period of less than ~20 years in Alaska and elsewhere. Comparing DEMs acquired at the same time of the year using the same radar wavelength is one promising strategy to limit the bias due to differential radar penetration (e.g., Neckel et al., 2013).

# Acknowledgements

We thank Tobias Bolch (editor), Robert McNabb and Mauri Pelto for their comments that greatly improved our manuscript. We thank Fanny Brun for sharing her python codes. We thank the Global Land Ice Measurement from Space (GLIMS) project that allowed the population of a vast archive of ASTER stereo images over glaciers. E. Berthier acknowledges support from the French Space Agency (CNES) and the Programme National de Télédétection Spatiale grant PNTS-2016-01.

## Author contributions

E.B. designed the study, made the data analysis and lead the writing. C.L. provided the laser altimetry data. W.D., M.W. and M.P. provided unpublished results. All authors discussed the results and wrote the paper.

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
