# Peer review of "Brief communication: Unabated wastage of the Juneau and Stikine icefields"

_The Cryosphere, 2017_

## Referee Comment (RC1) · R. McNabb (Referee) · 8 Feb 2018

**Summary**

In this manuscript, the authors have investigated the source of an apparent slowdown in the mass loss of the Juneau and Stikine Icefields, Alaska, by comparing multiple studies and data sources, re-processing the data used in a consistent fashion. They contend that the source of the signal seen is due to the use of the SRTM C-band DEM by two studies, Melkonian and others (2014) and Melkonian and others (2016), and not due to an actual slowing in the rate of mass loss for the glaciers studied. The

authors show that the unknown penetration depth of the SRTM C-Band radar signal into snow and ice causes a significant underestimation of elevation lowering, and therefore volume and mass losses for the two icefields. I think that the methods described in the manuscript are sound, and the results well-presented and reasonable. As such, I have only minor comments on the manuscript, otherwise recommending that it be accepted for publication in The Cryosphere.

**Minor comments**

- **l. 85**: Does this mean less than 0.5% of the icefield, after processing the DEMs and masking clouds, blunders, etc.?

- **l. 88**: Why the RGI v5.0, rather than v6.0?

- **l. 132**: Make sure the minus sign is on the same line as the number.

- **l. 134**: It's not clear to me what you mean by "statistically different for the JIF" - can you elaborate on this?

- **l. 181**: "for both datatsets"

- **Table 1**: It might be good to plot these data, perhaps as a supplemental figure, to ease the comparison of the values and mesh with your opening discussion statement.

- **l. 271**: I'm not sure what this sentence is meant to be saying - it seems like you stopped mid-thought while writing it.

---

## Referee Comment (RC2) · M. Pelto (Referee) · 24 Feb 2018

Berthier et al (2017) provide a detailed update of the mass balance record of two large icefields in southeast Alaska and an updated method of determination. The geodetic mass balance based on ASTER images is a significant improvement over previous combined analysis use ASTER and SRTM. Particularly on the Juneau Icefield the record is validated against both field mass balance observations and laser altimetry. The results is a robust record. The paper also provides a detailed review of how the SRTM data proved unreliable due to variable C-band penetration of snow and firn. This is an important and concise update in approach that should be used to reassess other

geodetic mass balance records that used best practices at the time, but may have this same correctable bias.

Specific Comments: The two different line numbers result from two different line numbering versions of the paper. Not sure which the authors will utilize.

40 or 45: Indicate that Lemon Creek Glacier is a WGMS reference glacier. Also separately note that the record to reflect that the Lemon Creek Glacier record from 2000-2016 is -0.56 m w.e. a-1 (WGMS, 2017).

65: Field observations of the ELA and mass balance do not support a slow down.

108 or 118: Taku Glacier (southern outlet of JIF)"

110 or 120: Should be reported that the most extensive thinning of the lower reach of JI glaciers is associated with lacustrine calving retreats on Field, Gilkey, Llewellyn, Meade, Mendenhall, and Tulsequah glacier (Pelto, 2017), which also notes that every outlet glacier retreated significantly except Taku Glacier. This supports the line 48 statement as well. On Stikine Icefield lacustrine and tidewater calving retreat during the study period occurred on Baird, Dawes, Great, Sawyer, South Sawyer, Speel and Wright Glacier.

146 or 156: Any thoughts on why the difference? This is in the terminus region for many glaciers including lake formation zone.

181 or 195: Separately note that the mass balance of Taku Glacier from 2000-2016 is -0.08 m w.e. a-1 (WGMS, 2017). Consider the value of citing Pelto et al (2008) pointed out the mass balance transition."Surface mass balance was positive from 1946-1988 +0.42 ma-1. This led to glacier thickening. From 1988-2006 an important change has occurred and annual balance has been -0.14 ma-1, and the glacier thickness has ceased increasing along Profile IV."

Table 1: Should add column for the field observed mass balance for Taku Glacier and Lemon Creek Glacier.

242 or 264: The linear correction used by Larsen et al (2007) would depend on the season of comparsion.

249 or 271: remove "is linked to the"

254 or 276: Winter balance observations on Taku Glacier support this seasonal amplitude.

262 or 286: which is in agreement with the altimetry and field based assessments on a smaller sample of these glaciers.

275 or 301: Is it worth elaborating for one sentence on the Tandem X issues? Also are the issues much reduced in summer for Tandem X?

Pelto, M. S., Miller, M. M., Adema, G. W., Beedle, M. J., McGee, S. R., Sprenke, K. F., and Lang, M.: The equilibrium flow and mass balance of the Taku Glacier, Alaska 1950-2006, The Cryosphere, 2, 147-157, https://doi.org/10.5194/tc-2-147-2008, 2008.

Pelto, M.: Recent Climate Change Impacts on Mountain Glaciers (John Wiley Sons, Inc.), chap. 3., 2017.

WGMS: Fluctuations of Glaciers Database. World Glacier Monitoring Service, Zurich, Switzerland. DOI:10.5904/wgms-fog-2017-10. Online access: http://dx.doi.org/10.5904/wgms-fog-2017-10, 2017.

---

## Author Response (AR1)

Etienne Berthier
av Ed Belin
31400 Toulouse
etienne.berthier@legos.obs-mip.fr

March 2018

Dear editor,

Please, find enclosed a revised version of our manuscript (MS) entitled "Brief communication: Unabated wastage of the Juneau and Stikine icefields (southeast Alaska) in the early 21st century". To facilitate your assessment, we uploaded a track-change version of the revised MS.

We thank the two reviewers for their positive evaluation of our study. Find below a copy of their comments and, in bold/blue, a point-by-point response to them. The revised text is provided in italics.

In addition to the reviewer's comments, we also corrected Figure 2b, where Melkonian et al. 2014 was replaced by Melkonian et al. 2016. We thank F. Brun for spotting this error.

We hope that these corrections/clarifications make our paper now suitable for publication in The Cryosphere.

Yours sincerely,

Etienne Berthier and co-authors

**Reply to reviewer#1, Robert McNabb**

Summary

In this manuscript, the authors have investigated the source of an apparent slowdown in the mass loss of the Juneau and Stikine Icefields, Alaska, by comparing multiple studies and data sources, re-processing the data used in a consistent fashion. They contend that the source of the signal seen is due to the use of the SRTM C-band DEM by two studies, Melkonian and others (2014) and Melkonian and others (2016), and not due to an actual slowing in the rate of mass loss for the glaciers studied. The authors show that the unknown penetration depth of the SRTM C-Band radar signal into snow and ice causes a significant underestimation of elevation lowering, and therefore volume and mass losses for the two icefields. I think that the methods described in the manuscript are sound, and the results well-presented and reasonable. As such, I have only minor comments on the manuscript, otherwise recommending that it be accepted for publication in The Cryosphere.

Minor comments l. 85: Does this mean less than 0.5% of the icefield, after processing the DEMs and masking clouds, blunders, etc.?

> **Reply: The sentence has been clarified and now reads "***Images in which valid elevation data covered less than 0.5% of the icefield areas were excluded, ...***"**

l. 88: Why the RGI v5.0, rather than v6.0?

> **Reply: At the time of our study, RGI v6.0 was not published. Anyway, outlines are unchanged for the study area between RGI 5.0 and 6.0.**

l. 132: Make sure the minus sign is on the same line as the number.

> **Reply: Thanks, we will be careful while proof-reading the article.**

l. 134: It's not clear to me what you mean by "statistically different for the JIF" - can you elaborate on this?

> **Reply: We improved the text by adding "***, i.e. the JIF mass balances do not overlap given the error bars.***"**

l. 181: "for both datatsets"

> **Reply: Corrected here (and elsewhere in the MS).**

Table 1: It might be good to plot these data, perhaps as a supplemental figure, to ease the comparison of the values and mesh with your opening discussion statement.

> **Reply: Supplementary Figure S2 added and referred to in the opening discussion statement**

l. 271: I'm not sure what this sentence is meant to be saying - it seems like you stopped mid-thought while writing it.

> **Reply: Sorry of the typo. "is linked to the" has now been removed.**

**Reply to reviewer#2, Mauri Pelto**

Berthier et al (2017) provide a detailed update of the mass balance record of two large icefields in southeast Alaska and an updated method of determination. The geodetic mass balance based on ASTER images is a significant improvement over previous combined analysis use ASTER and SRTM. Particularly on the Juneau Icefield the record is validated against both field mass balance observations and laser altimetry. The results is a robust record. The paper also provides a detailed review of how the SRTM data proved unreliable due to variable C-band penetration of snow and firn. This is an important and concise update in approach that should be used to reassess other geodetic mass balance records that used best practices at the time, but may have this same correctable bias.

Specific Comments: The two different line numbers result from two different line numbering versions of the paper. Not sure which the authors will utilize.

or 45: Indicate that Lemon Creek Glacier is a WGMS reference glacier. Also separately note that the record to reflect that the Lemon Creek Glacier record from 2000-2016 is -0.56 m w.e. a-1 (WGMS, 2017).

> **Reply: We now indicate in the introduction that Lemon Creek is a WGMS reference glacier. The 2000-2016 mass balance of this glacier is also compared to our ASTER-based estimates at the end of the Results section.**

65: Field observations of the ELA and mass balance do not support a slow down.

> **Reply: Statement added in the introduction. "***Field observations of the equilibrium line altitude and mass balance do not support a slow down (WGMS, 2017).***"**

or 118: Taku Glacier (southern outlet of JIF)"

> **Reply: text corrected.**

or 120: Should be reported that the most extensive thinning of the lower reach of JI glaciers is associated with lacustrine calving retreats on Field, Gilkey, Llewellyn, Meade, Mendenhall, and Tulsequah glacier (Pelto, 2017), which also notes that every outlet glacier retreated significantly except Taku Glacier. This supports the line 48 statement as well. On Stikine Icefield lacustrine and tidewater calving retreat during the study period occurred on Baird, Dawes, Great, Sawyer, South Sawyer, Speel and Wright Glacier.

> **Reply: We believe that this is beyond the scope of the paper. Our goal is not to compare and try to explain the variability in individual glacier mass balances. Together with the reference noted by the reviewer, the issue has been addressed in detail in Larsen et al., GRL, 2015. The text is thus unaltered.**

or 156: Any thoughts on why the difference? This is in the terminus region for many glaciers including lake formation zone.

> **Reply: We have no straightforward explanation for these differences. No such difference is observed for the terminus area of the Stikine icefield. We further note that the fraction of the glacier area covered by these terminus regions are rather small so that they do not count much in the overall mass budget. However, we reckon that they are important for process understanding.**

or 195: Separately note that the mass balance of Taku Glacier from 2000-2016 is -0.08 m w.e. a-1 (WGMS, 2017).

**Reply: Text has been modified to include the 2000-2016 Taku mass balance.**

Consider the value of citing Pelto et al (2008) pointed out the mass balance transition."Surface mass balance was positive from 1946-1988 +0.42 ma-1. This led to glacier thickening. From 1988-2006 an important change has occurred and annual balance has been -0.14 ma-1, and the glacier thickness has ceased increasing along Profile IV."

**Reply: This reference was added in the introduction. "***This statement is also valid for Taku Glacier, for which the mass balance was positive (+0.42 m w.e. a-1) from 1946-1988 and negative (-0.14 m w.e. a-1) during 1988-2006 (Pelto et al., 2008).***"**

Table 1: Should add column for the field observed mass balance for Taku Glacier and Lemon Creek Glacier.

**Reply: As glaciological mass balance are available for only two of the compared glaciers we prefer to mention them in the discussion text, not in the table.**

or 264: The linear correction used by Larsen et al (2007) would depend on the season of comparison.

**Reply: We are unsure what the referee means here. Our text is thus currently unaltered.**

or 271: remove "is linked to the"

**Reply: Removed. Thanks for spotting this.**

or 276: Winter balance observations on Taku Glacier support this seasonal amplitude.

**Reply: We modified the text, stating:** *"Assuming a seasonal amplitude as large as 10 m (a value in agreement with field measurements of the Juneau Icefield Mass Balance Program, Pelto et al., 2013), the slope of the regression"*

or 286: which is in agreement with the altimetry and field based assessments on a smaller sample of these glaciers.

**Reply: Statement added in the first sentence of the introduction.**

or 301: Is it worth elaborating for one sentence on the Tandem X issues? Also are the issues much reduced in summer for Tandem X?

[revised manuscript text omitted]

---

## Author Response (AR2)

Etienne Berthier
av Ed Belin
31400 Toulouse
etienne.berthier@legos.obs-mip.fr

April 2018

Dear Tobias,

Please, find enclosed a revised version of our manuscript (MS) entitled "Brief communication: Unabated wastage of the Juneau and Stikine icefields (southeast Alaska) in the early twenty-first century". To facilitate your assessment, we uploaded a track-change version of the revised MS.

We thank you for your careful reading of the paper and your comments. Find below a copy of them and, in bold/blue, a point-by-point response. The revised text is provided in italics.

We hope that these corrections/clarifications make our paper now suitable for publication in The Cryosphere.

Yours sincerely,

Etienne Berthier and co-authors

**Reply to the Editor, Tobias Bolch**

Dear Etienne, dear co-authors,

I carefully read the manuscript and your reply to the comments. The comments were mainly of minor nature. Most of them have been addressed, but I would have expected slightly more careful incorporation. Some sentences were added as a response to the comments, but they partly do not fit into the flow of the text, e.g. line 41: It is not clear to which statement you are referring to. Lemon Creek Glacier had a negative balance throughout while the balance for Taku Glacier was clearly positive for a period.

> **Reply: We changed "The statement is" to "***The trend toward enhanced mass loss***"**

L. 61: Be more specific about field observations. Which glacier(s) were observed? Entire Stikine icefield?

> **Reply: We changed the sentence to "***Field observations of the equilibrium line altitudes and surface mass balances on Lemon Creek and Taku glaciers (JIF) also do not support a slowdown (WGMS, 2017).***"**

L. 85 Images do not cover elevation data. One can guess what you mean, but it is not clear from the text.

> **Reply: "Images" replaced by "***DEMs***"**

Please check once again all added statements and sentences carefully.

> **Reply: Checked everywhere**

In addition, I do not agree to not consider a comment because it is not easy to understand. You can at least provide a guess or better write an email to the reviewer who provided his name to ask for clarification.

> **Reply: We assume that the editor refers here to the following comment "The linear correction used by Larsen et al (2007) would depend on the season of comparison". We contacted directly Mauri Pelto and he told us by email "This was more in support of what you were saying. The seasonal correction just adds an uncertainty/error to their determination that your methodology does not. I was suggesting, maybe not so clearly, that you could add this as a supportive/explaining comment. There was not a question to respond to."**

> **The linear correction was calculated between two dataset with a clear timestamp (February 2000 and August 2000) so it is not clear to us why the correction would depend on the season of comparison. It was an empirical way for Larsen et al. (2007) to make the data seasonally compatible by adjusting the SRTM DEM to a summer elevation dataset.**

> **No change was made to the manuscript.**

It is also a valid comment to mention that extensive thinning of some glaciers is due to calving events. In case you do not want to include a third study by the reviewer (which I would understand) you can refer the Larsen et al. (2015) as mention in the reply.

> **Reply: It is not a problem of adding another reference. It is simply that our study is not about understanding the cause of glacier loss in southeast Alaska and neither about the drivers of variability between individual glaciers. In fact, our results do not bring any new**

**insights on the processes governing glacier mass loss so we strongly believe that there is no point in mentioning the extensive thinning at lake-terminating (not tidewater) glaciers. To avoid any ambiguity/disappointment for the reader, we clarify this by adding the statement: "***Understanding the pattern of dh/dt and its variability among glaciers is beyond the scope of this brief communication and the reader is referred to earlier publications on this topic (e.g., Larsen et al., 2015).***"**

In addition, I have two more substantial comments and several minor specific comments:

1. I asked you to provide more information about the utilized glacier outlines. Years were added, but ask you to carefully check once again. As I understood (not entirely clear from Kienholz et al. 2015) for some regions outlines from Bolch et al. (2010) were used and some images were not acquired in 2004/05. In addition, and as a good example as your papers are highly recognized, you should cite the original source of the outlines if possible.

**Reply: we contacted Christian Kienholz who told us:**

**"The Bolch et al. (2010) outlines were used for the Canadian/eastern part of the JIF, as stated in Figure 2c of the Kienholz et al. (2015) paper. However, the outlines may not be fully identical to the original outlines. For example, we used ALOS-derived streamlines to check/update the divides across the entire JIF (see Figure 3a in my 2015 paper). Adding a sentence that the RGI outlines are based on outlines compiled by Bolch et al. (2010) and Kienholz et al. (2015) may be good, as indicated by Tobias. Also, Figure 2 from my 2015 paper should be added to the next RGI technical guide to avoid confusion.**

**2004/2005 is correct for the bulk of the glaciers across the Juneau and Stikine Icefields. A few glaciers were updated using imagery from 2010/2011."**

**We also checked the RGI technical guide about recommandation for referencing RGI. The recommendation is that "The RGI may be used freely with due acknowledgement (by citing this note for technical details or Pfeffer et al. 2014 for scientific background)". Because we wanted to recognize the tedious work that accompanies the making of such a detail inventory in Alaska (in line with your comments), we made sure we cited the original reference. So we checked the Alaska chapter in the he RGI technical guide which mentions "Changes from Version 3.2 to 4.0. A new inventory compiled by C. Kienholz (Kienholz et al., submitted), including topographic and hypsometric attributes, replaces the former inventory of Alaska". Kienholz et al. was thus cited in our paper. Based on our personal communication with C. Kienholz, we now also cited the Bolch et al. 2010 study in our revised paper.**

**You and Christian Kienholz were deeply involved in the RGI. If the recommendation provided on the technical guide (just citing the technical guide + Pfeffer et al. 2014) does not suit you, then I think there should be an open discussion among the RGI leaders about it. As users of these outlines, we are afraid that finding/citing all the sources that were compiled in the RGI would become a tedious work and we feel that it was not really the spirit of the RGI.**

**We also checked again the dates of individual glacier outlines in the RGI attributes and found only image dates in 2004 and 2005. Despite the statement by C. Kienholz that a few glaciers were updated using more recent imagery. The revised text is:**

*"The RGI v5.0 glacier outlines for both the JIF and SIF were mapped using imagery acquired in majority in August of 2004 and 2005 (Bolch et al., 2010; Kienholz et al., 2015). "*

But more important you are analysing the period 2000 – 2016, but the outlines are from 2004/05. Hence, a justification why you use outlines which do not match the investigation period is needed. Or were they adjusted?

> **Reply: The editor is right, glacier outlines were not adjusted to the start/end dates of your geodetic mass balance estimate. This important information was indeed missing in our paper. A statement is added in the revised text and the negligible effect on the mass balance is supported by a sensitivity analysis for the Northern Patagonia Icefield (3800 km²) in Dussaillant et al., 2018: "***No updated inventory is available or was produced in this study for the JIF and SIF. Therefore, we neglected changes in glacierized area between 2000 and 2016, and assumed that mass balance uncertainties linked to area changes are covered by our 5% area uncertainty (Paul et al., 2013; Dussaillant et al., 2018)****."**

2. I agree that the comparison of DEM differencing results to repeat laser altimetry is not straight forward as you mention in L. 170. However, this is not only because of the different time periods. This is also due to different coverage. I did not read the paper in detail, but as I understood from Larson et al. (2015), mainly glacier centrelines were measured. Hence, the mass loss might be overestimated when scaling to the entire glacier in case no correction is included as mentioned by Berthier et al. (2010), NatGeo. This issue needs to be tackled and discussed.

> **Reply: Good point. We now added a paragraph about this in the comparison of the mass balances using the two techniques. "***A further complication for the comparison of our ASTER-based results to repeat laser altimetry arises from different spatial sampling: mostly continuous coverage from DEMs vs. centreline sampling from laser altimetry. Berthier et al. (2010) found that centreline sampling could lead to an overestimation of mass loss. In their study, two large and rapidly retreating glaciers (Bering and Columbia, outside of our study domain) were responsible for 92% of the overestimation of the mass loss from centreline profiling (Table S4 in Berthier et al., 2010). Overestimation was not obvious for other glaciers. More recently, Johnson et al. (2013) presented an improved treatment of laser altimetry data and found no such overestimation from centerline profiling over the Glacier Bay region (southeast Alaska). In their improved processing, each change in elevation (dz) is assigned to a mid-point between old and new elevations whereas in the original laser altimetry analysis (Arendt et al., 2002), dz were assigned to the old elevation..****"**

> **And also in the discussion: "***This agreement suggests that an appropriate analysis of centreline data may be sufficient to measure the glacier-wide mass balance of these glaciers as previously shown for the nearby Glacier Bay area (Johnson et al., 2013). .****"**

Specific comments:

L. 12/17. It is a matter of style, but I would not use abbreviations in the abstract, if not really needed to save words.

> **Reply: abbreviations removed. Good point.**

L. 16: remove ","

> **Reply: removed.**

L. 52: Where did you get the information about the mass balances from? Include a reference.

> **Reply: ref to (Larsen et al., 2015) repeated. It was not clear indeed.**

L. 62: I'd omit the word "further".

> **Reply: omitted.**

L. 66-70. This statement with more or less similar wording is repeated in L. 239ff. You may once again refer to the problem of the x-band radar penetration but with a different wording. But more important, you need to be more specific about the x-band penetration (under which conditions can the penetration be so high?) and not just provide a general statement. In case you are at the word limit avoid the repetition but provide this relevant information instead.

> **Reply: we find it difficult to avoid the repetition and think it is helpful for the reader to know right away in the introduction how Melkonian et al. addressed the penetration issue. It will help to understand how we designed our study and why revisiting the ASTER analysis is needed. It is maybe not so problematic to repeat twice that recent studies have found clear penetration of the X-Band signal into cold snow and firn at a time when many colleagues are using Tandem-X data for geodetic mass balance estimates? We fully agree with the editor that it is indeed important to add that such high penetration depth is observed under specific conditions and we now write: "*X-band penetration depth has recently been recognized to reach several meters in cold and dry snow/firn*".**

L. 93: According to my knowledge the automatically generated ASTER DEMs which are available are called "AST14DEM". The "AST14DMO" includes both the DEM and the orthoimage generated using this DEM.

> **Reply: true. Thanks. Changed everywhere and also in the color code of Figure 2.**

L. 95: Co-registration is crucial. Hence include a short statement with reference regarding the co-registration.

> **Reply: We stated a few line above "Planimetric and altimetric offsets of each ASTER DEM were corrected using the SRTM DEM as a reference". We now added a reference to Nuth and Kääb (2011).**

L. 112: Include one/two sentences how the uncertainties where calculated and then refer to the reference for more details.

> **Reply: More details about this uncertainty assessment is given now.**

L. 192: Check sentence. Write glaciological mass balance (also L. 196), so that it fully clear that these are values are based on the glaciological method.

> **Reply: Sentence corrected.**

L. 196: write "was" instead of "is".

> **Reply: corrected.**

Table 1: Add uncertainty rages.

> **Reply: Added.**

L. 252: Repetition of "consider"

> **Reply: corrected.**

L. 294: I think you can make an even broader statement here as the penetration might also be underestimated in several other studies.

> **Reply: we followed this suggestion but the broader statement was included a few lines further down the text "*Caution should thus be used when deriving mass balance using SRTM and Tandem-X DEMs over time period of less than ~20 years in Alaska and elsewhere*".**

L. 297: I think it is very crucial to be more precise of the x-band penetration. The penetration depth depends also on the depth of the snow and firn layers

**Reply: we added "***under cold and dry conditions***".**

L. 301: I'd move this important statement to the discussion and put slightly more emphasis on it.

**Reply: we prefer to keep it here at the end of the conclusion as this is not a result from out study but a perspective on how to use more safely the tandem-X DEM.**

I am looking forward to your revised version. Please include a reply to each comment and highlight the changes made in to manuscript.

Do not hesitate to ask in case you have a question.

Best regards,

Tobias - Editor TC

[revised manuscript text omitted]